# Below Average Cognitive Ability—An under Researched Risk Factor for Emotional-Behavioural Difficulties in Childhood

**DOI:** 10.3390/ijerph182412923

**Published:** 2021-12-08

**Authors:** Andrea K. Bowe, Anthony Staines, Deirdre M. Murray

**Affiliations:** 1INFANT Research Centre, Department of Paediatrics, University College Cork, T12 K8AF Cork, Ireland; d.murray@ucc.ie; 2School of Nursing, Psychotherapy and Community Health, Dublin City University, 9 Dublin, Ireland; Anthony.staines@dcu.ie; 3Department of Paediatrics and Child Health, Cork University Maternity Hospital, Wilton, T12 DC4A Cork, Ireland

**Keywords:** early intervention, cognitive ability, emotional-behavioral function

## Abstract

Children with below average cognitive ability represent a substantial yet under-researched population for whom cognitive and social demands, which increase in complexity year by year, may pose significant challenges. This observational study examines the longitudinal relationship between early cognitive ability and emotional-behavioral difficulties (EBDs) between the age of three and nine. Participants include 7134 children from the population-based cohort study growing up in Ireland. Cognitive ability was measured at age three using the Picture Similarities Scale. A t-score one to two standard deviations below the mean was defined as below average cognitive ability (*n* = 767). EBDs were measured using the Strengths and Difficulties Questionnaire (SDQ) at three, five, and nine years of age. Generalized linear mixed models and logistic regression were used to examine the relationship. Below average cognitive ability was an independent predictor of higher longitudinal SDQ scores. After adjustment, children with below average cognitive ability were 1.39 times more likely (AOR 1.39, 95% CI 1.17–1.66, *p* < 0.001) to experience a clinically significant EBD between the ages of three to nine years. This study demonstrates the increased risk of EBDs for children with below average cognitive ability. A scalable method of early identification of at-risk children should be a research priority for public health, enabling early intervention for cognitive and adaptive outcomes.

## 1. Introduction

Children with below average cognitive ability represent a substantial proportion of children, and often have unrecognized difficulties [1,2]. Intellectual function and adaptive behavior that are approximately two or more standard deviations (SD) below the mean on a standardized test are diagnostic of an intellectual developmental disability (IDD) [3]. Those who perform one to two SD below the mean fall into a “grey area”, which is substantially below that of the average population but not meeting the diagnostic criteria for IDD.

Early deficits in cognitive ability are associated with later adverse outcomes in many domains of life including mental and physical health, educational attainment, socio-economic status, and substance use [4,5,6,7]. Despite well-established adverse impacts, approximately 13% of children whose cognitive ability falls into this “grey area” receive little attention either in the scientific literature or in ordinary life.

Unlike children with more significant IDDs, whose marked deficits are likely to be detected on routine developmental screening, children with below average cognitive ability often reach their early motor and language milestones, and routinely commence formal education without recognition of the difficulties they may face and without additional educational supports [8,9,10]. This has potential to adversely affect both their transition into school, and their early years of experience in education, a critical period in the development of social and emotional well-being [11,12,13].

The phrase “the first 1000 days”, a period spanning conception to three years of age, is one increasingly seen in policy documents relating to child health [1]. The phrase has its origins in the work of professor David Barker, a physician and epidemiologist, whose work prompted the development of the ‘fetal origins hypothesis’, also known as “Barker’s hypothesis” [14,15]. A fundamental concept in Barker’s work was developmental plasticity, which he described as ‘a critical period when a system is plastic and sensitive to the environment, followed by loss of plasticity, and a fixed functional capacity’ [6]. With regard to cognitive development specifically, this concept refers to periods during which neuronal connections are acutely susceptible to environment-dependent modification, and after which function capacity remains fixed [15]. The “critical period” concept has since been challenged and while it is now broadly accepted that neuronal circuits can adapt throughout life, it is also accepted that early life is a unique period, differing from adulthood, during which the brain shows greater potentiality toward plasticity [16]. Since Barker, intervention in early life has been examined in lab-based studies, human intervention studies, and in epidemiological studies. There is now consensus agreement across the literature that early life is a period of fundamental importance in cognitive development, making it the ideal target for early intervention programs.

The challenge for children with below average cognitive ability is that the early years of education may not unearth the cognitive difficulties experienced by children with below average ability. Thereafter, the increasing complexity of cognitive tasks and the accelerated academic demand may uncover underlying difficulties for these children, who may struggle to keep pace with their typically developing peers [17]. Unfortunately, a child may have to consistently fail the tasks asked of them for many years before they are considered for formal educational assessment [10]. The implications of this repeated failure on their psychological outcomes, particularly on emotional-behavioral development, is unclear.

EBDs are more prevalent in children with intellectual disabilities, though the literature examining the larger proportion of children with below average cognitive ability specifically is limited. There are few population-based studies which have examined this relationship. Studies which have focused specifically on those with below average or borderline intellectual ability suggest that these children are at increased risk of EBDs compared to their typically developing peers, though gaps exist in the literature in relation to the change in emotional-behavioral function over time (and its temporal relationship with cognitive ability and commencing formal education) [17,18,19,20]. Real-time recognition of below average cognitive ability occurs rarely [8,10]. For this reason, epidemiological studies of population-based data are ideally placed to identify the difficulties experienced by this substantial proportion of children, to inform policy and service planning, and to adequately meet their educational and psychological needs.

The aim of this study, which is based on a nationally representative sample of children in Ireland, is to inform early childhood policy on early intervention through a detailed examination of the impacts of below average cognitive function on emotional-behavioral well-being.

The hypothesis underlying this study is that emotional-behavioral trajectories will differ between those with and without below average cognitive ability, particularly around the time of entry to formal education, and that children identified as below average cognitive ability will be at increased risk of clinically significant EBDs in childhood.

The objectives of this study are: (1) to describe the epidemiology of below average cognitive ability at age three (2) to model emotional-behavioural trajectories between the age of three to nine years for those with and without below average cognitive ability; and (3) to examine the independent association between early childhood cognitive ability and clinically significant EBDs in childhood adjusting for relevant confounding factors.

## 2. Materials and Methods

### 2.1. Study Design and Setting

This study is based on secondary analysis of data from the Growing up in Ireland (GUI) Infant ’08 Cohort, a nationally representative longitudinal cohort study of infants in Ireland, designed to inform government policy in relation to children, young people, and families [21]. The GUI dataset contains information collected from parents, teachers, school principals, and the children themselves. Information was collected on diverse aspects of the child’s early life including birth characteristics, parental health, family income, dietary and exercise habits, as well as outcomes including physical health and development, social emotional and behavioral well-being, and educational achievement and intellectual ability.

### 2.2. Participants

Families with infants born between December 2007 and May 2008 were identified from the Child Benefit Register, a register used to administer universal child benefit payments in Ireland. Potential participants were stratified according to marital status of claimant, county of residence, nationality, and number of children in the claim, before a systematic selection procedure with a random start and constant sampling fraction were applied. Full details of the GUI study are available elsewhere [21]. Data collection for Wave 1 began in September 2008 and recruited 11,134 nine-month-old infants and their primary caregivers, representing a 65% response rate. The participation and non-response rates for each wave of data collection are described in Appendix A.

This study is based on data from 7134 infants and their primary caregivers who completed Wave 1 (nine months), Wave 2 including cognitive assessment (age three), Wave 3 (age five) and Wave 5 (age nine), representing 67.4% of the original 11,134 participants. Questionnaires were administered to the primary caregiver in their home using computer assisted personal interviewing.

### 2.3. Ethical Approval

Ethical approval for the GUI study was obtained from the Research Ethics Committee, Department of Children and Youth Affairs Ireland. Written consent was collected from parents/guardians prior to data collection. Secondary analysis of the anonymized Microdata Files of the GUI study does not require additional ethical approval. The Strengthening the Reporting of Observational Studies in Epidemiology (STROBE) guidelines were followed in conducting and reporting this study [22].

### 2.4. Measures

#### 2.4.1. Cognitive Ability

Non-verbal cognitive ability was measured at age three using the Pictures Similarities Scale, a cognitive ability sub-test from the British Abilities Scales (BAS) which measures non-verbal reasoning [23]. The BAS has demonstrated construct validity and reliability as a measure of intellectual ability [24,25]. In a time-restricted survey setting the BAS is particularly beneficial as each core sub-test is individually interpretable [23].

The test was administered by an interviewer at the home of the child. The child was shown four images and was given a card containing a single fifth image. The child must match the card to one of the four pictures based on a shared characteristic. Provided in the dataset were raw scores, ability scores, and standardized scores. The standardized scores adjust for the performance of the child compared with other children of the same age [23,25]. The mean of the standardized score for the Picture Similarities Scale was 52.60 with a SD of 10.78. Those with scores < 2 SD below the mean, consistent with a possible diagnosis of an intellectual disability, were excluded from the main analyses (*n* = 216), but a description of this cohort along with the results of models including this cohort are included in Appendix A (Appendix A, Appendix A) for information. Those with a score one to two SD below the mean were categorized as below average cognitive ability and those with a score greater than 1 SD below the mean as average/above cognitive ability.

#### 2.4.2. Emotional-Behavioral Development

Emotional-behavioral development was measured at age three, age five, and age nine using the parent-reported version of the Strengths and Difficulties Questionnaire (SDQ). The SDQ is comprised of four difficulties scales measuring emotional symptoms, conduct problems, hyperactivity/inattention, and peer relationship problems. Validation of the SDQ is well documented [26,27]. An overall difficulty score for each child was calculated by combining the score of these four scales. The Total SDQ score was treated as both a continuous and categorical variable. In line with previous literature, scores above a 90th percentile cut-off were defined as “clinically significant”. This corresponded to a cut off of ≥14 at age three, ≥14 at age five, and ≥16 at age nine.

#### 2.4.3. Confounding Variables

Confounding variables were chosen a priori based on previously published literature demonstrating a relationship with both the predictor and outcome of interest. All were measured at the baseline survey at nine months by questionnaire completed by the primary caregiver. For 99∙7% of participants the primary caregiver was the mother.

Child characteristics included sex (male, female), gestational age (weeks), birth weight (<2500 g, 2500–4000 g, >4000 g), and admission to Neonatal Intensive Care Unit at birth (yes, no). Maternal characteristics included maternal age and maternal depression score (eight-item Center for Epidemiologic Studies Depression Scale). Socio-demographic characteristics included partner in household (yes, no); highest level of maternal education (lower secondary or less, upper secondary, sub degree (includes certificates and diplomas) and third level (primary degree and above); and income quintile (five equally sized groups each containing 20% of participants based on ‘equivalized’ disposable family income taking account of household size, composition, and disposable income after statutory deductions).

Where potential multicollinearity arose (maternal education & household income quintile; gestational age, birthweight, and admission to NICU) each variable was trialed in the model and chosen based on best model fit, assessed using the Bayesian Information Criterion (BIC).

### 2.5. Statistical Analysis

Statistical analysis was conducted using ‘R Statistical Software Package’ [28]. A weighting variable calculated by the GUI study was applied to the population of 7134 participants to adjust the structure of the sample to align with the population as a whole, allowing inferences to be made about the population from which participants were sampled [26]. Participants with complete data on the variables of interest were included in the analyses. Missing data was minimal and was ≤2% for all variables included in the linear and logistic analyses (Appendix A).

Descriptive statistics were used to characterize the study population according to child, maternal, and sociodemographic characteristics. Pearsons chi-square tests and students t-tests were used to examine for significant differences in characteristics between cognitive ability groups. Correlations between the non-verbal cognitive ability score at age three and Total and Subscale SDQ scores at age three, five, and nine were examined using Pearson’s correlation coefficient and are described in Appendix A.

The outcome of interest was total SDQ score at age three, five, and nine. Generalized linear mixed models [29,30] were used to examine the relationship between early cognitive ability and sequentially obtained total SDQ scores (main effect), and to examine the relationship between early cognitive ability and the change in SDQ during childhood (interaction effect). Between participant variability in baseline SDQ was modelled using a random intercept and between participant variability in change of SDQ over time was modelled using a random slope.

The relationship was first examined using univariable analyses. Multivariable models were then built using a hierarchical approach, with the addition of infant characteristics, maternal characteristics, and sociodemographic characteristics. Model fit was assessed following addition of each set of variables using the BIC, with lower values indicating better model fit.

To evaluate the relationship between cognitive ability and the change in SDQ during childhood an interaction term between cognitive ability and age at SDQ measurement was investigated. Age was modelled as both a continuous and categorical variable, with the latter displaying better model fit. Each confounding variable was entered into an interaction term with age and if statistically significant on univariable analysis was then included in the multivariable model. Significant interaction terms with child, maternal, and sociodemographic characteristics were then added in blocks as before.

The independent association between early cognitive ability and the presence of a clinically significant EBD at age three, five, and nine was examined using logistic regression.

## 3. Results

### 3.1. Description of the Sample

There were 7507 participants who completed the waves of interest in the GUI study. There were *n* = 157 participants with missing data on the Picture Similarities test at age three and *n* = 216 participants with a score <2 SD below the mean who were excluded. The population of 7134 participants are described in Table 1 with respect to their cognitive ability at age three. At age three, 10.8% of the study population were categorized as below average cognitive ability. The remaining 89.2% were categorized as average/above.

### 3.2. Characteristics of Study Population

Mothers of children with below average cognitive ability at age three were significantly more likely to report no partner present in household (21.4% v 14.7%, *p* < 0.001), and to experience higher mean depression scores (3.00 +/− 4.20 vs. 2.46 +/− 3.60, *p* < 0.001). There were statistically significant differences between the cognitive groups in relation to maternal education, income, and household social class. In the below average group 31.0% of mothers reported their highest level of education to be lower secondary or less compared with 17.4% in the average or above group. Similarly, 28.3% were in the lowest income quintile and 15.8% in the lowest social class group, compared to 19.0% and 8.7%, respectively, in the average/above group. Males comprised 61.8% of the below average group.

### 3.3. Total SDQ Scores

The total SDQ scores for each cognitive ability group at age three, five, and nine are shown in Figure 1. There were statistically significant differences in mean total SDQ scores between cognitive ability groups at each age, with the below average cognitive group having a higher mean total SDQ at each time point (age three 8.21 vs. 7.36 *p* < 0.001, age three 7.80 vs. 6.88 *p* < 0.001, age three 8.23 vs. 6.83 *p* < 0.001). To visualize the individual SDQ pathways, total SDQ scores at each age were divided into deciles with decile 1 representing the lowest 10% of scores and decile 10 the highest 10% of scores, and an alluvial plot was created for each cognitive ability group (Appendix A). At each age a higher proportion of the below average cognitive ability group had total SDQ scores in the highest deciles, and a higher proportion remained in the highest deciles throughout childhood.

### 3.4. Cognitive Ability as a Predictor of Longitudinal Total SDQ Scores

The predictive significance of early cognitive ability was examined using four models, described in Table 2. In Model 1, a univariable model, early cognitive ability was a significant predictor of higher longitudinal SDQ scores (BIC 120,291). Model 2 adjusts for child characteristics (BIC 120,231, *p* < 0.001). Model 3 adjusts for child and maternal characteristics (BIC of 119,628, *p* < 0.001), and Model 4 for child, maternal, and sociodemographic characteristics (BIC of 119,508, *p* < 0.001). Cognitive ability remained a significant predictor of higher longitudinal total SDQ scores. Parameter estimates for the fully adjusted Model 4 are summarized in Figure 2.

### 3.5. Cognitive Ability as a Predictor of Change in Total SDQ Score

The change in mean SDQ score during childhood differed according to cognitive ability (F = 5.330, *p* = 0.005). The predicted change in Total SDQ between age three and nine for each cognitive ability group is shown in Figure 3. The unadjusted and adjusted interaction models are described in Appendix A.

### 3.6. Cognitive Ability and Clinically Significant Emotional Behavioural Difficulties

The proportion of those with a clinically significant EBD remained at 10% between age three and nine in the average/above cognitive ability group. Among those with below average cognitive ability, the proportion increased from 14.5% at age three, to 15.9% at age five, and 16.5% at age nine. The unadjusted and adjusted logistic regression models examining the relationship between below average cognitive ability and the presence of a clinically significant emotional-behavioral difficulty are described in Table 3. After adjusting for relevant confounding variables, compared to their peers with average/above ability, those with below average ability were 1.4 times more likely to experience a clinically significant EBD between age three and nine (AOR 1.39 95% CI 1.17–1.66, *p* < 0.001).

## 4. Discussion

We have shown, in a large and well characterized prospective cohort study of more than 7000 children, that the mean trajectory of change in emotional-behavioral development between age three and nine differs for those with below average cognitive ability.

The first objective of the study was to describe the epidemiology of below average cognitive ability in early childhood. At age three, 10∙8% of children scored one to two SD below the mean on the Picture Similarities Subtest of the British Ability Scales. There was no significant difference between cognitive groups in relation to potential biological risk factors including birthweight, gestational age, or admission to a neonatal intensive care unit at birth. This finding differentiates these children from those with more severe non-genetic intellectual disabilities for whom low birthweight and early gestational age are well established risk factors [31]. The sociodemographic characteristics of the groups differed significantly, with the below average group significantly more likely to be characterized by low maternal education, low income, and single parent households. There is a substantial body of literature demonstrating the profound effect adverse socioeconomic environments have on infant and child neurodevelopment [32,33,34]. The exact mechanistic pathway is complex and likely mediated by multiple factors including prenatal influences, parental care, and cognitive stimulation [34].

The second objective of the study was to model emotional-behavioral trajectories between age three and nine for those with and without below average cognitive ability. The a priori hypothesis was that children with below average cognitive ability would experience a different trajectory of change in emotional-behavioral function between age three and nine compared to their counterparts with average/above cognitive function. We postulated that the time of commencing formal education, which typically occurs at age four to five, would signal a divergence between the groups and those with below average cognitive ability would experience increasing difficulties in emotional-behavioral function. The results support this hypothesis. Unlike their counterparts who experienced an overall decrease in mean SDQ score between age three and nine, those with below average cognitive ability experienced a mean increase, with the vast majority of this increase occurring between the age of five and nine.

The multivariate logistic regression model with stringent adjustment confirmed that below average cognitive ability increased the odds of having a clinically significant EBD between the age of three and nine. Of note, when the model was fitted for each age point separately there was a non-significant association with EBD at age three. However, at age five and age nine, after formal education has commenced, lower cognitive ability was associated with significantly increased odds of EBD.

There is limited published literature examining this specific cohort, but the findings of our study appear consistent with what has been published to date. To our knowledge there is one published population-based study which has focused specifically on childhood mental health and borderline intellectual functioning, a level of intellectual function defined by an IQ between 70–85, a construct similar to that of below average cognitive ability. This 2010 study which was based on more than 4300 participants in the Longitudinal Study of Australian Children (LSAC), 12.1% of whom were categorized as borderline intellectual functioning at age four to five, reported the prevalence of possible mental health problems at age six to seven years in this cohort was 17% compared to 5% among typically developing peers. After controlling for potential confounding due to socioeconomic disadvantage using propensity score matching, significant differences in prevalence remained between those with limited intellectual ability and those typically developing [19].

A 2019 study using the same population cohort but at later waves of the study examined this association in the adolescent period and reported children with borderline intellectual functioning at age eight to nine had four times the odds (adjusted odds ratio 4.33, 95% confidence interval 2.84–6.62) of having abnormal emotional-behavioral difficulties scores at age 14/15 years [18]. Neither study examined the change in emotional-behavioral function over time for these groups.

Disentangling the relationship between cognitive ability and emotional-behavioral development is complex. Firstly, there is overlap between the constructs making it difficult to sharply distinguish between certain elements of both abilities. For example, attention is a component of both the broad construct of cognitive function, but also of behavior [35,36]. Secondly, there are many confounding variables in the relationship between cognitive ability and emotional-behavioral development, as risk factors such as childhood adversity and markers of lower socioeconomic status predispose to both outcomes.

Acknowledging these challenges in establishing a causal relationship between cognitive ability and emotional-behavioral development, the results of this study suggest a worsening in emotional-behavioral function for those with below average cognitive ability between the age of five and nine, a period coinciding with increasingly complex cognitive tasks as one progresses through formal education.

The findings of this study are subject to limitations, perhaps the most significant being the use of a single performance metric as a measure of cognitive ability in childhood. Use of this single measure, while accepted as being individually interpretable within the British Ability Scales, does not account for alternative, yet equally important abilities, such as musical or artistic abilities, nor does it account for a child’s adaptive functioning in the presence of lower cognitive ability. Additionally, cognitive ability is not static construct and will change over time. Further research using methods to account for this change may help to further decipher the temporal relationship between cognitive ability and emotional-behavioral function.

A further limitation of the study is that it is based on a longitudinal survey and is therefore subject to selection bias. Those who participate in population-based studies, particularly those who remain engaged in longitudinal studies, are likely to systematically differ from those who do not across key socio-demographic domains. To account for this, to the extent possible, the weighting variable calculated by the GUI study, to ensure the longitudinal cohort were representative of the population from which they were drawn, was applied.

## 5. Conclusions

The relationship between early cognitive ability and emotional-behavioral function in childhood has not to-date been extensively investigated. While there are a number of systematic reviews and meta-analyses reporting a disproportionately high prevalence of mental health problems among children with intellectual developmental disabilities, no such reviews exist for the larger proportion of children with below average cognitive ability.

In this study we examined the relationship between early cognitive ability and emotional-behavioral development using a variety of statistical models. We found that below average cognitive ability was associated with higher longitudinal SDQ scores in childhood, and that children with early below average cognitive ability are at higher risk of clinically significant emotional-behavioral difficulties in childhood. We also demonstrated that the change in emotional-behavioral function over time differs according to cognitive ability and that for those with below average cognitive ability, the period between the age of five and nine signals a worsening in emotional-behavioral function.

To our knowledge, this is the first study to examine in a longitudinal manner the change in mean SDQ over time for such a cohort of children. The findings of this study are suggestive of a temporal relationship between commencing formal education and worsening emotional behavioral function for this cohort of children. It is entirely plausible that this period of increasingly complex cognitive and social demand, during which one is routinely compared to typically developing peers, may culminate in the manifestation of emotional or behavioral problems. For many, it is only at this stage they will begin the process of assessment and support. Wieland and Zitman have outlined why taking this approach of waiting for adaptive difficulties to develop before recognizing or intervening is problematic, and how for many it results in “problematic lives, functioning under high strain but unnoticed by the rest of society” [8].

This research raises the question of how early childhood policies can be adapted to

Identify high risk children;Provide effective early intervention to address both cognitive and emotional-behavioral needs.

The early identification of children at risk of below average cognitive ability remains an important area for further research. The risk factors for a child failing to meet their full cognitive potential have been extensively researched and appear rooted in the social determinants of health. Factors such as parental education, family income, socioeconomic status have shown consistent and strong associations with low cognitive ability. However, routinely this information is not utilized in early life to identify high risk children.

In addition, there is a paucity of research examining interventions to improve emotional-behavioral and cognitive outcomes specifically for those with below average cognitive ability, who are likely to differ in many ways from those with more severe intellectual disabilities (with whom they are often categorized).

The key question which remains, is how as a society, health, and social care system, we can recognize and support these children to help them to reach their full potential.

## Figures and Tables

**Figure 1 ijerph-18-12923-f001:**
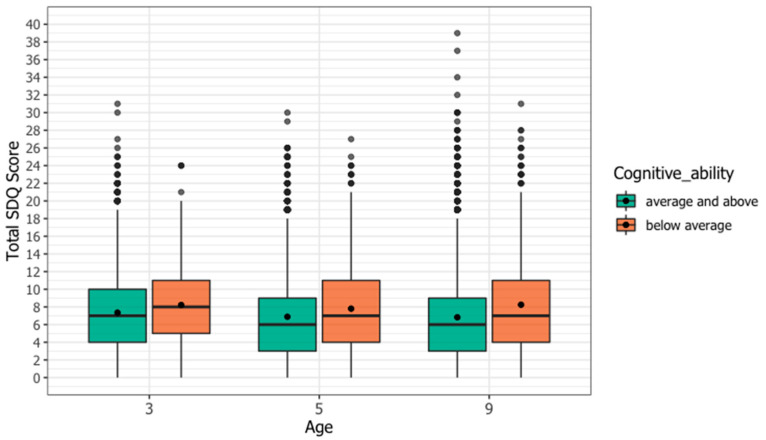
Distribution of total SDQ Scores at age three, age five, and age nine by cognitive ability group. Distribution of total SDQ Scores at age three, age five, and age nine by cognitive ability group. The distribution of scores at each age is shown in a boxplot. Each box represents the 25th to 75th percentile. The length of the box is the interquartile range (IQR). The horizontal line inside the box represents the median. The whiskers represent the lowest and highest quartiles. The dot inside the box represents the weighted mean. Circles represent individual outliers. At each age, those with below average cognitive ability had higher mean and median total SDQ scores than those with average/above cognitive ability.

**Figure 2 ijerph-18-12923-f002:**
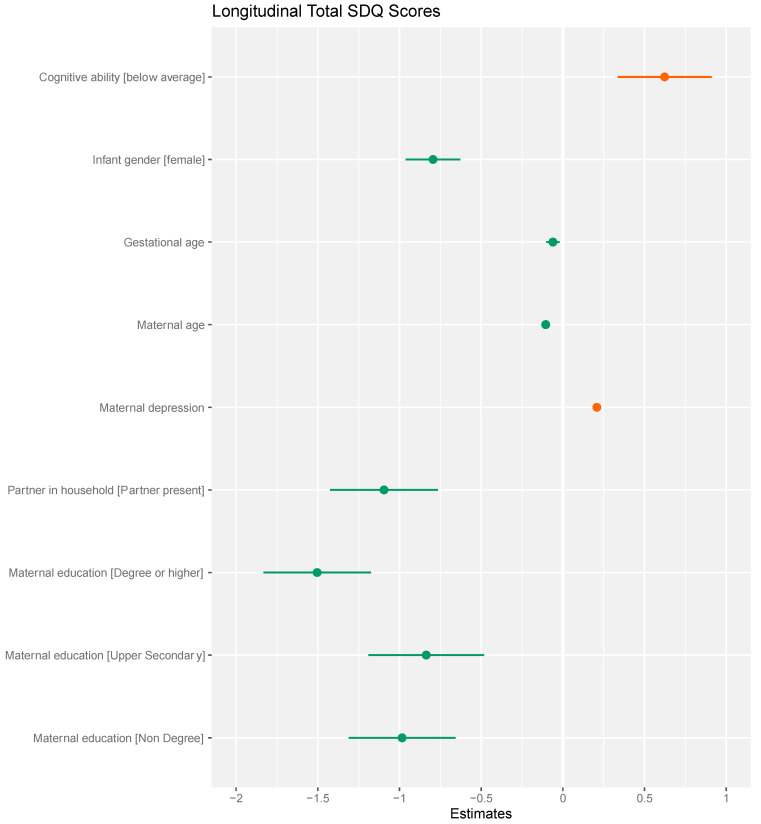
Parameter estimates for general linear model 4 examining relationship between below average cognitive ability and longitudinal SDQ scores. Parameter estimates of generalized linear mixed model investigating predictive significance of cognitive ability for longitudinal total SDQ scores. This plot demonstrates the parameter estimates for the variable of interest, below average cognitive ability, and for the included confounding variables. Having below average cognitive ability was associated with higher longitudinal total SDQ scores. Reference Categories: Infant gender—Male, Partner in household—Partner absent, Maternal education—Lower secondary.

**Figure 3 ijerph-18-12923-f003:**
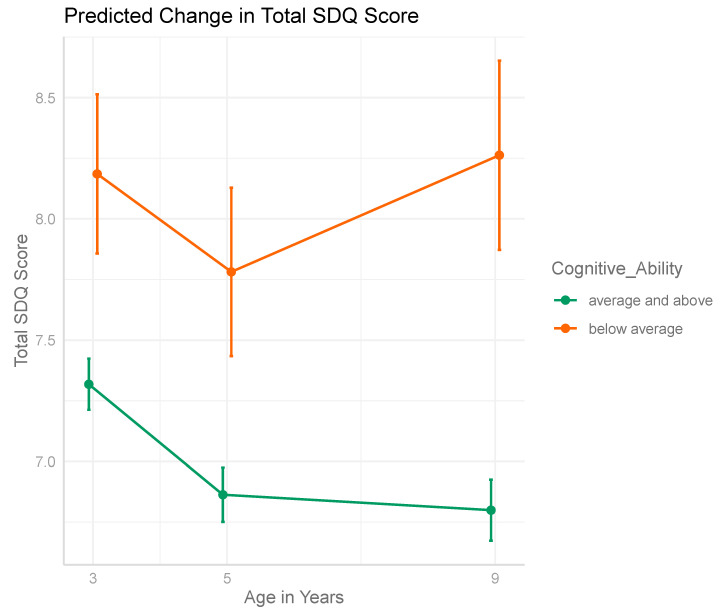
Predicted change in total SDQ score between age three and nine by cognitive ability group.

**Table 1 ijerph-18-12923-t001:** Characteristics of study population according to cognitive ability at age three.

			Average/Above Cognitive Ability	Below Average Cognitive Ability	
	Valid	Total	*n* (%)	*n* (%)	*p*-Value
			6362 (89.2)	772 (10.8)	
**Maternal Characteristics**					
Age in first year of life *	7134				
N, Mean (sd)		7134	6362, 31.6 (5.3)	772, 31.0 (5.7)	0.002
Partner in household in first year	7134				
No		1099	934 (14.7)	165 (21.4)	<0.001
Current Smoker	7134				
Yes		1917	1642 (25.8)	275 (35.6)	<0.001
Depression					
N, Mean (SD)	7030		6276, 2.46 (3.60)	754, 3.00 (4.20)	<0.001
**Sociodemographic Characteristics**					
Maternal Education	7132				
Lower Secondary		1343	1104 (17.4)	239 (31.0)	
Upper Secondary		1758	1547 (24.3)	211 (27.3)	
Sub Degree		2053	1877 (29.5)	176 (22.8)	
Third Level and higher		1978	1832 (28.8)	146 (18.9)	<0.001
Income Quintile	6576				
Lowest		1315	1113 (19.0)	202 (28.3)	
2nd		1314	1150 (19.6)	164 (22.9)	
3rd		1344	1195 (20.4)	149 (20.8)	
4th		1398	1286 (21.9)	112 (15.7)	
Highest		1205	1117 (19.1)	88 (12.3)	<0.001
Household Social Class	6431				
Professional/Managerial		3391	3107 (49.0)	284 (37.1)	
Non-Manual/Skilled Manual		2364	2108 (33.3)	256 (33.4)	
Semi-skilled/Unskilled Manual		676	571 (9.0)	105 (13.7)	
Never Worked—no class defined		672	551 (8.7)	121 (15.8)	<0.001
**Infant Characteristics**					
Gender	7134				
Male		3617	3140 (49.4)	477 (61.8)	
Female		3517	3222 (50.6)	295 (38.2)	<0.001
Birth weight	7062				
<2500 g		408	354 (5.6)	54 (7.0)	
2500–4000 g		5655	5049 (80.2)	606 (79.0)	
>4000 g		999	892 (14.2)	107 (14.0)	0.283
Gestational age in weeks ^	7119				
N, mean (sd)			6347, 39.5 (2.1)	772, 39.5 (2.3)	0.400
Admission to NICU at birth	7127				
Yes		976	857 (13.5)	119 (15.4)	0.141

* Maternal age includes *n* = 532 participants aged 40 or older who were coded as age 40 in original dataset. ^^^ Gestational age includes *n* = 10 participants with a gestational age of 25 weeks or less was recorded as 25.

**Table 2 ijerph-18-12923-t002:** Results of generalized linear mixed model examining cognitive ability as a predictor of longitudinal SDQ scores.

	Model 1 Main Effect Cognitive Ability	Model 2 Adjusted for Child Factors	Model 3 Adjusted for Child and Maternal Factors	Model 4 Adjusted for Child, Maternal, and Sociodemographic Factors
Intercept	7.07 (6.98–7.17)	9.76 (8.04–11.47)	14.34 (12.57–16.12)	14.91 (13.13–16.69)
Below average cognitive ability	1.01 (0.70–1.31)	0.91 (0.61–1.21)	0.78 (0.49–1.07)	0.62 (0.34–0.91)
Infant Gender: Female		−0.77 (−0.94–−0.59)	−0.77 (−0.94–−0.60)	−0.79 (−0.96–−0.63)
Gestational age (weeks)		−0.06 (−0.10–−0.01)	−0.07 (−0.11–−0.03)	−0.06 (−0.10–−0.02)
Maternal age			−0.14 (−0.16–−0.13)	−0.10 (−0.12–−0.09)
Maternal depression score			0.23 (0.20–0.25)	0.21 (0.18–0.23)
Partner present				−1.09 (−1.42–−0.76)
Education: Upper Secondary				−0.84 (−1.19–−0.48)
Education: Non-Degree				−0.98 (−1.31–−0.66)
Education: Degree or higher				−1.50 (−1.83–−1.18)
Participant number	7079	7079	7079	7079
Observations	21,237	21,237	21,237	21,237
Marginal/Conditional R2	0.005/0.911	0.013/0.911	0.077/0.914	0.092/0.919
BIC	120,291.3	120,231.4	119,628.6	119,508.3
*p*-value		<0.001	<0.001	<0.001

BIC: Bayesian Information Criterion.

**Table 3 ijerph-18-12923-t003:** Logistic regression model examining odds of a clinically significant emotional-behavioral difficulty (EBD) between age three and nine.

	Any EBD		Unadjusted Models		Adjusted Model *		
	No	Yes	Crude OR	95% CI	AOR	95% CI	*p*-Value
Cognitive Ability							
Average and above	4866 (78.5)	1330 (21.5)	Ref.		Ref.		
Below Average	505 (67.8)	240 (32.2)	1.72	1.46–2.03	1.39	1.17–1.66	<0.001
Gender:							
Male	2626 (74.5)	898 (25.5)	Ref.		Ref.		
Female	2745 (80.3)	673 (19.7)	0.69	0.62–0.77	0.70	0.62–0.78	<0.001
Depression Score, n, mean (sd)	5611, 2.13 (3.26)	1754, 3.81 (4.58)	1.11	1.10–1.13	1.09	1.07–1.10	<0.001
Education							
Lower Secondary ^	870 (64.1)	460 (35.9)	Ref.		Ref.		
Upper Secondary	1260 (73.4)	457 (26.6)	0.67	0.58–0.78	0.74	0.63–0.87	<0.001
Non-Degree	1627 (81.2)	377 (18.8)	0.44	0.38–0.51	0.54	0.46–0.65	<0.001
Degree or higher	1665 (85.8)	276 (14.2)	0.30	0.26–0.36	0.45	0.37–0.54	<0.001
Gestational age n, mean (sd)	5686, 39.57 (2.00)	1774, 39.37 (2.40)	0.96	0.94–0.98	0.96	0.94–0.99	0.004
Partner							
No	617 (57.5)	456 (42.5)	Ref.		Ref.		
Yes	4754 (81.0)	1115 (19.0)	3.04	2.67–3.46	0.54	0.46–0.64	<0.001
Maternal Age Category							
≤24 years	497 (58.5)	353 (41.5)	Ref.		Ref.		
25–34 years	3058 (78.8)	821 (21.2)	0.37	0.32–0.43	0.64	0.53–0.76	<0.001
≥35 years	1817 (82.1)	397 (17.9)	0.30	0.26–0.36	0.54	0.45–0.66	<0.001

OR: odds ratio; AOR: adjusted odds ratio; 95% CI: 95% confidence interval; * Adjusted for partner, highest level education, maternal age category, maternal depression Score, gender of child, and gestational age in weeks. ^^^ Refers to lower secondary education or less.

## Data Availability

The anonymized microdata file of the Growing Up in Ireland study was accessed for this study from the Irish Social Science Data Archive (ISSDA)—www.ucd.ie/issda, (accessed on 8 July 2021). This data is available for bona fide research purposes upon written request and approval from ISSDA.

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
