# Peer review of "Below Average Cognitive Ability—An under Researched Risk Factor for Emotional-Behavioural Difficulties in Childhood"

_ijerph, 2021, doi:10.3390/ijerph182412923_

Round 1

Reviewer 1 Report

Please refer to the appended file.

Reviewer 2 Report

This an exceptionally well written study that presents secondary data of a long-term observational study (GUI) on the relationship between early cognitive impairment and later emotional deficits. The study thereby adds to a growing clinical interest to diagnose and treat pre-school children. 

My only real concern is the use of a single metric measurement of Cognitive Ability as also acknowledged by the authors. Particularly, the analyses relied solely on a sub-scale of the British Abilities Scale (BAS), which clearly is unfavorable from a psychometric perspective, even if the BAS battery is considered to yield individually interpretable sub-test results. Taking also into account that diagnostic instruments for pre-school children are in general far less reliable and valid compared to older age groups (e.g., Scheeringa & Haslett, 2010) I feel that the study benefits from additional discussion:

  1. The authors should help clarify the relative independence of “cognitive ability” and “emotional-behavioural difficulties”. Are these constructs sharply distinct, overlapping, or naturally confounded (for example “hyperactivity/inattention” is considered as a criterion for the latter [line 127], but attention-direction is equally known as a cognitive skill)? Is it possible to present further statistical analysis (e.g. correlations) or theoretical evidence from the GUI data-set or corresponding literature to tackle this measurement issue? If not, this should be explicitly stated.
  2. The presumed causal interplay between low cognitive ability and emotional-behavioural difficulties that the statistical evidence apparently demonstrates could be further illuminated. The authors may therefore consider interpreting results against hierarchical information processing models, e.g. by Nfer & Nelson (1997), which has been priorly linked to discussions of the BAS (Hill, 2005).
  3. The authors may also report how the targeted group of “underachievers” (i.e. 1-2 SD below the BAS mean) differ from clinically disabled children (i.e. <2SD below the BAS mean)? Likely, results of those 216 children hold relevant information and warrant presentation as supplementary data.

Such additional discussion would allow to draw clearer implications for diagnostics and early intervention, hence support the studies' ecological validity, further.
